# An Account of Models of Molecular Circuits for Associative Learning with Reinforcement Effect and Forced Dissociation

**DOI:** 10.3390/s22155907

**Published:** 2022-08-07

**Authors:** Zonglun Li, Alya Fattah, Peter Timashev, Alexey Zaikin

**Affiliations:** 1Department of Mathematics, University College London, London WC1E 6BT, UK; 2Institute for Women’s Health, University College London, London WC1E 6BT, UK; 3World-Class Research Center “Digital Biodesign and Personalized Healthcare”, Sechenov University, Moscow 119991, Russia; 4Department of Applied Mathematics and Laboratory of Systems Biology of Aging, Lobachevsky State University of Nizhny Novgorod, Nizhny Novgorod 603022, Russia

**Keywords:** associative learning, molecular circuits, synthetic biology, mathematical modelling, Hill equation, Pavlov’s dog, reinforcement, dissociation, nondimensionalisation

## Abstract

The development of synthetic biology has enabled massive progress in biotechnology and in approaching research questions from a brand-new perspective. In particular, the design and study of gene regulatory networks in vitro, in vivo, and in silico have played an increasingly indispensable role in understanding and controlling biological phenomena. Among them, it is of great interest to understand how associative learning is formed at the molecular circuit level. Mathematical models are increasingly used to predict the behaviours of molecular circuits. Fernando’s model, which is one of the first works in this line of research using the Hill equation, attempted to design a synthetic circuit that mimics Hebbian learning in a neural network architecture. In this article, we carry out indepth computational analysis of the model and demonstrate that the reinforcement effect can be achieved by choosing the proper parameter values. We also construct a novel circuit that can demonstrate forced dissociation, which was not observed in Fernando’s model. Our work can be readily used as reference for synthetic biologists who consider implementing circuits of this kind in biological systems.

## 1. Introduction

Synthetic biology is an emerging field that involves re-engineering existing biological systems or creating new ones that may solve real-world problems in medicine, agriculture, etc. [1,2,3,4,5,6,7,8,9]. Over the past few decades, synthetic biology has witnessed a rapid revolution in the biotechnology industry, and opened up enormous potential for next-generation research in biology due to the increasingly tremendous power of genetic engineering technology, and ever decreasing cost of synthesis and sequencing [10,11,12,13,14,15,16,17]. Particularly, increasing attention has been drawn to designing and testing synthetic biological circuits in vitro, in vivo and in silico in an attempt to better understand bioartificial intelligence at the cellular and molecular levels [18,19,20,21,22,23]. These artificial circuits can, therefore, function as fundamental units to modify existing cellular behaviours, and to perform a wide range of tasks of our own interest in programmable organisms [24,25,26,27,28]. Numerous synthetic circuits were developed for associative learning, decision making, and oscillators, and a brief summary is exhibited in Table 1 [29,30,31,32,33,34,35,36,37]. With the growing collaboration between theorists and experimentalists in almost every discipline, we also noticed a trend in synthetic biology that mathematical models are frequently used to acquire insight, inform troubleshooting, and perform predictions [38,39,40,41].

Associative learning occurs in many aspects of our life, and it is regarded as the basis of our understanding of other forms of behaviours and cognition in human and nonhuman animals [42,43,44,45,46]. The most classical experiment on associative learning is Pavlov’s dog, in which the dog associated the ring of a bell with the smell of food [47,48]. The dog learned to associate the conditioned stimulus (bell’s ring) with the unconditioned stimulus (smell), such that next time, in the presence of the bell’s ring alone, the dog knew that the food would be served soon, and the learned response (the saliva from its mouth) was observed. The historical viewpoint is that the mammalian nervous system plays a vital role in associative learning through neuronal signaling and reconfiguration [49,50,51,52].
sensors-22-05907-t001_Table 1Table 1Types of synthetic circuits.Associative LearningDecision MakingOscillatorFernando et al., 2009 [53]Nene et al., 2012 [30]Stricker et al., 2008 [36]Nesbeth et al., 2016 [18]Filicheva et al., 2016 [31]Tigges et al., 2009 [37]Macia et al., 2017 [34]Abrego and Zaikin, 2017 [32]Borg et al., 2022 [35]

However, some studies revealed the possibility that non-neural agents may also organise in a similar fashion [34,54,55]. Naturally, molecular circuits may display similar behaviours as molecular reactions form the building block of cellular activities. As a result, the design and investigation of molecular interactions that manifest associative learning have become active research topics. Although logic gates have been widely adopted in synthetic biology for emulating diverse biological behaviours [56,57,58,59,60], in this article, the focus is confined to the continuous models constructed by the Hill equation [61,62], since continuous models tend to generate more accurate results and facilitate the comprehension of fine details of the system. One of the first models is Fernando’s model, where the authors designed a genetic circuit that mimicked Hebbian learning in a neural network architecture [53]. That work stands as a well-organised interdisciplinary article in which a mathematical model was developed, and the biotechnological approach that can be implemented in *Escherichia coli* is thoroughly discussed. Nonetheless, because of its interdisciplinary nature, the work only demonstrates the fact that learning can be formed after conditioning, but fails to investigate other, more advanced behaviours, such as the reinforcement effect and forced dissociation. More specifically, it is natural to assume that the learned response becomes stronger with increasing times of conditioning. For illustrative brevity, we name the phenomenon *reinforcement effect*. It is also reasonable to suppose that the response becomes weaker with the repeated cuing of conditioned stimulus (with no unconditioned stimulus taking place at the same time) shortly after the formation of conditioning, and we call it *forced dissociation*. To the best of our knowledge, neither of them has been formally discussed in the previous literature. The potential importance of these two behaviours can be explained by two aspects. On the one hand, in Pavlov’s dog experiment, the former would be equivalent to the scenario that the repeated conditioning of the bell and the food would reinforce the dog’s belief that the bell’s ring is a reminder of food availability; the latter would be that the repeated bell’s ring alone shortly after the conditioning would stop the dog reckoning that the bell is related to the food’s availability. On the other hand, these properties may provide experimentalists with more flexibility over the control of some biological systems, as synthetic circuits are widely used to regulate them [4,6,63,64]. We show that Fernando’s model is able to manifest the reinforcement effect by choosing the proper parameters, but unable to manifest forced dissociation. This motivated us to design a new circuit that possesses the potential to display forced dissociation. The new circuit also involves fewer proteins and does not contain any feedback loop, which can potentially reduce the wiring complexity in practical implementation. In the meantime, we also study the robustness of respective models to the Hill coefficients, as this instructs experimentalists on the type of polymers that can be used to implement the circuit.

## 2. Models

### 2.1. Fernando’s Model

The circuit diagram of Fernando’s model is shown in Figure 1, and we assumed that the circuit could be implemented in a programmable cell. Unlike the schematic diagram given in [53], here we omit genes for illustrative simplicity. In the diagram, each oval box denotes a particular protein. The activation was drawn with an arrow, and the inhibition was drawn with a hammerhead. Except for inhibitions $u_{1}\to r_{1}$ and $u_{2}\to r_{2}$, where the input molecules were directly bound to the repressors, all other activations and inhibitions were realised by transcription and translation. For instance, repressor $r_{1}$ inhibits the transcription of a particular gene, which guides the manufacture of molecule $\omega_{1}$. The upstream gene of protein *p* has two available operator sites, one for $r_{1}$, $\omega_{1}$ and another for $r_{2}$, $\omega_{2}$. A more detailed explanation can be found in [53]. Fernando’s model is characterised by the system as follows (N=2):(1)$$\left\{\begin{array}{cc} \frac{dp}{dt} & =\sum_{j=1}^{N}v_{p}\left(\frac{\omega_{j}^{a}}{K_{\omega }^{a}+\omega_{j}^{a}}\right)\left(\frac{K_{r}^{b}}{K_{r}^{b}+r_{j}^{b}}\right)-\delta_{p}p\\ \frac{d\omega_{j}}{dt} & =v_{\omega }\left(\frac{p^{b}}{K_{p}^{b}+p^{b}}\right)\left(\frac{K_{r}^{b}}{K_{r}^{b}+r_{j}^{b}}\right)-\delta_{\omega }\omega_{j}+\epsilon_{j}\\ r_{j} & =\frac{R}{1+ku_{j}} \end{array}\right.$$

In the equations above, $u_{1}$ and $u_{2}$ represent the respective concentrations of the unconditioned and conditioned stimuli, and they are given to the cell in a transient time window. $\omega_{1}$ and $\omega_{2}$ represent the respective concentrations of the weight molecules. $r_{1}$ and $r_{2}$ represent the respective concentrations of the repressor molecules. The concentration of the response molecule is denoted by *p*. $K_{\omega }$, $K_{r}$ and $K_{p}$ denote the respective Hill constants for molecule ω, *r* and *p* that measure the concentrations of the transcription factors required for half occupancy. *R* denotes the repressor concentration in the absence of molecule *u*. *a* and *b* are Hill coefficients that measure the cooperativity of the respective transcriptional factors. In [53], the authors used a=4 and b=2; however, in this work, we study the impact of varying integer values of *a* and *b* on qualitative behaviours. ϵ denotes the basal grow rate, and we assumed that it was only nonzero for j=1. *v* and δ denote the growth and degradation rate parameter, respectively, and the subscripts are used to signify the source of contribution. The architecture shows that the genetic circuit was structurally symmetric, and the left and the right halves were independent. The association was triggered by the feedback of response molecule *p*, and the inspiration came from Hebbian learning, which dictates information exchange between neurons [65].

In order for an association to be formed, we can simply render the concentration of the molecule $\omega_{1}$ abundant, and molecule $\omega_{2}$ insignificant before the start of the experiment. When only molecule $u_{1}$ was given to the cell, it was bound to repressor molecule $r_{1}$ and reduced the concentration of molecule $r_{1}$. Therefore, the inhibition of the transcription with respect to the genes controlled by molecule $r_{1}$ was lifted. Eventually, sufficient molecule $\omega_{1}$ activated the transcription with respect to the gene associated with response molecule *p* and promoted the production of *p*. Conversely, when only molecule $u_{2}$ was given to the cell, we could not observe abundant molecule *p* due to the shortage of $\omega_{2}$ availability. However, at the time when molecule $u_{1}$ was paired with molecule $u_{2}$, the production of molecule *p* (triggered by $u_{1}$) elevated the concentration of molecule $\omega_{2}$ because of the feedback loop, so that the next time, even when only molecule $u_{2}$ was present, there already existed a sufficient molecule $\omega_{2}$ for the production of *p*, which implies that the association was formed.

In order to analyse a system of differential equations, one often converts the system to a dimensionless scale as a first step. One generally reduces the volume of parameters and removes physical units from the system, which facilitates mathematical investigations and renders the model more flexible for experimentalists who wish to implement the system in vivo or in vitro, as the units are not specified. By using scaling $\bar{\omega_{j}}=\frac{\omega_{j}}{K_{\omega }}$, $\bar{r_{j}}=\frac{r_{j}}{K_{r}}$, $\bar{p}=\frac{p}{K_{p}}$, $\bar{t}=\delta_{p}t$, $\bar{u_{j}}=ku_{j}$, the dimensionless model becomes (overlines were dropped for simplicity):(2)$$\left\{\begin{array}{cc} \frac{dp}{dt} & =\sum_{j=1}^{N}\alpha \left(\frac{\omega_{j}^{a}}{1+\omega_{j}^{a}}\right)\left(\frac{1}{1+r_{j}^{b}}\right)-p\\ \frac{d\omega_{j}}{dt} & =\beta \left(\frac{p^{b}}{1+p^{b}}\right)\left(\frac{1}{1+r_{j}^{b}}\right)-\theta \omega_{j}+\tau_{j}\\ r_{j} & =\frac{S}{1+u_{j}} \end{array}\right.$$
where $\alpha =\frac{v_{p}}{K_{p}\delta_{p}}$, $\beta =\frac{v_{\omega }}{K_{\omega }\delta_{p}}$, $\theta =\frac{\delta_{\omega }}{\delta_{p}}$, $\tau_{j}=\frac{\epsilon_{j}}{K_{\omega }\delta_{p}}$, $S=\frac{R}{K_{r}}$.

Now, we try to intuitively interpret whether the reinforcement effect can be realised by Fernando’s model or not. Suppose we carry out the conditioning twice in order to guarantee that the second learned response was more abundant than the first, the simplest way is to ensure that the initial concentration of $\omega_{2}$ is small, and the growth parameter β is not too large, so that we could anticipate *p* to keep growing with the repeated conditioning according to Equation (Equation 2). Additionally, we needed to select a relatively small θ to guarantee that the memory that was reflected by the weight molecules did not disappear too rapidly. Next, we investigate how the two stimuli dissociate. Intuitively, in light of the design of Fernando’s model, the disappearance of the learned response was dictated by the time elapse. This is due to the fact that $\omega_{2}$ promotes the production of *p*, such that the response eventually disappears only if $\omega_{2}$ falls to 0. Additionally, the learned response does not attenuate if the time interval between the two successive stimuli is insufficient relative to the decay rate. In all, the dissociation is autonomous and is not dictated by the repeated cuing of the conditioned stimulus (alone). This can be circumvented by a different design, which is introduced next.

### 2.2. A Model with Forced Dissociation

The circuit for the model proposed here was inspired by [18] and is shown in Figure 2. In the diagram, each oval box denotes a particular protein molecule, and we again omitted the genes for the illustrative brevity. More specifically, input *x* initiates the transcription of a particular gene which guides the manufacture of molecule *u*; the translation of molecule *v* is controlled by another gene, the expression of which is dictated by *x*, *u* and *z* together. Similarly, *y* is controlled by a third gene, the expression of which is dictated by *x*, *v* and *z*. The activation was drawn with an arrow, and the inhibition was drawn with a hammerhead. Input *x* plays the role of the conditioned stimulus, input *z* plays the role of the unconditioned stimulus, and *y* represents the response. However, the discrimination between the conditioned and unconditioned stimuli was downplayed in some sense; we explain this in the discussion. The design of this new architecture comes with at least two purposes. First, we intended to construct a simple circuit of which the mechanism was completely different from that of Fernando’s model, aiming for the components of the circuit to be more interactive with each other, and the association not to be reliant upon the participation of the feedback loops. These may lift a few restrictions on the synthetic implementation. Second, given that Fernando’s model is not capable of demonstrating forced dissociation, we hope to build a model that could successfully dissociate the two stimuli by repeating the cuing of the conditioned stimulus alone right after the conditioning. To fulfil the latter requirement, instead of placing a molecule ($\omega_{2}$ in Fernando’s model) that promoted the transcription of the response protein molecule, we could actually consider introducing an inhibitor upstream of the response molecule. In this way, the consistent input of conditioned stimulus alone was expected to reduce the amount of the response molecule, so long as the stimulus promoted the expression of the inhibitor after the conditioning of the conditioned and unconditioned stimuli. This explains why we introduce the x→v→y pathway in Figure 2. Regarding the other parts of the circuit, z→y and z→v→y guarantee that input *z* can always activate output *y*, x→y ensures that there can exist a sufficient learned response upon the formation of associative learning, and *u* functions as a moderator, speeding up the consumption of *v* that renders the reinforcement effect more likely to occur. Here, we assumed that the Hill coefficients for all molecules were identical, and they are denoted by *a*.

By employing the Hill equation, the system can be described with the following equations:(3)$$\left\{\begin{array}{cc} \frac{dy}{dt} & =\alpha_{yx}\left(\frac{x^{a}}{K_{x}^{a}+x^{a}}\right)\left(\frac{K_{z}^{a}}{K_{z}^{a}+z^{a}}\right)\left(\frac{K_{v}^{a}}{K_{v}^{a}+v^{a}}\right)\\ & +\alpha_{yz}\left(\frac{K_{x}^{a}}{K_{x}^{a}+x^{a}}\right)\left(\frac{z^{a}}{K_{z}^{a}+z^{a}}\right)\left(\frac{K_{v}^{a}}{K_{v}^{a}+v^{a}}\right)\\ & +\alpha_{xyz}\left(\frac{x^{a}}{K_{x}^{a}+x^{a}}\right)\left(\frac{z^{a}}{K_{z}^{a}+z^{a}}\right)\left(\frac{K_{v}^{a}}{K_{v}^{a}+v^{a}}\right)\\ & -\delta_{y}y\\ \frac{du}{dt} & =\alpha_{ux}\frac{x^{a}}{K_{x}^{a}+x^{a}}-\delta_{u}u\\ \frac{dv}{dt} & =\alpha_{vx}\left(\frac{x^{a}}{K_{x}^{a}+x^{a}}\right)\left(\frac{K_{u}^{a}}{K_{u}^{a}+u^{a}}\right)\left(\frac{K_{z}^{a}}{K_{z}^{a}+z^{a}}\right)-\delta_{v}v \end{array}\right.$$

Similarly to Fernando’s model, $K_{x}$, $K_{z}$, $K_{v}$, and $K_{u}$ denote the Hill constants for molecules *x*, *z*, *v*, and *u*, respectively. The production and degradation rates are denoted by α and δ, where the subscripts are used to signify the source of contribution. Then, we used a similar scaling approach to render the system dimensionless. The dimensionless system is shown below:(4)$$\left\{\begin{array}{cc} \frac{dy}{dt} & =\alpha_{yx}\left(\frac{x^{a}}{1+x^{a}}\right)\left(\frac{1}{1+z^{a}}\right)\left(\frac{1}{1+v^{a}}\right)\\ & +\alpha_{yz}\left(\frac{1}{1+x^{a}}\right)\left(\frac{z^{a}}{1+z^{a}}\right)\left(\frac{1}{1+v^{a}}\right)\\ & +\alpha_{xyz}\left(\frac{x^{a}}{1+x^{a}}\right)\left(\frac{z^{a}}{1+z^{a}}\right)\left(\frac{1}{1+v^{a}}\right)\\ & -y\\ \frac{du}{dt} & =\alpha_{ux}\frac{x^{a}}{1+x^{a}}-\beta_{u}u\\ \frac{dv}{dt} & =\alpha_{vx}\left(\frac{x^{a}}{1+x^{a}}\right)\left(\frac{1}{1+u^{a}}\right)\left(\frac{1}{1+z^{a}}\right)-\beta_{v}v \end{array}\right.$$

One of the drawbacks of this circuit is that the model is not as heuristic as Fernando’s model. Indeed, Fernando’s model borrows the architecture of Hebbian learning, whereas we built our model from scratch, tailored for the properties that we wanted to achieve. Intuitively, in order for the reinforcement effect to occur, we anticipated that, by properly choosing the parameter values, the presence of *x* alone could significantly elevate the amount of *v*, and the conditioning (whenever both *x* and *z* are present) could speed up the consumption of *v*. This ensured that the second learned response is more abundant than the first, and the response to the conditioned stimulus alone was simultaneously less abundant than that to the unconditioned stimulus. When it comes to forced dissociation, as we discussed previously, the repeated cuing of input *x* alone after the conditioning could elevate the concentration of *v*, which subsequently reduces output *y* as a result of x→v→y. The numerical result is given in the next section.

Before closing this section, we briefly mention a potential adjustment to the current circuit that could simplify the system given in Equation (Equation 4). Our existing scheme allows for *x*, *v*, and *z* to bind at a single operator site. In fact, we can adjust the output part in a way such that *z* exploits operator sites that only restrict to *z* itself, which render it looking somehow analogous to the output part of Fernando’s model, in which the left half and the right half are unrelated. The dimensionless model arising from the adjustment can then be reduced to: (5)$$\left\{\begin{array}{cc} \frac{dy}{dt} & =\alpha_{yx}\left(\frac{x^{a}}{1+x^{a}}\right)\left(\frac{1}{1+v^{a}}\right)+\alpha_{yz}\left(\frac{z^{a}}{1+z^{a}}\right)-y\\ \frac{du}{dt} & =\alpha_{ux}\frac{x^{a}}{1+x^{a}}-\beta_{u}u\\ \frac{dv}{dt} & =\alpha_{vx}\left(\frac{x^{a}}{1+x^{a}}\right)\left(\frac{1}{1+u^{a}}\right)\left(\frac{1}{1+z^{a}}\right)-\beta_{v}v \end{array}\right.$$

The adjustment provides an alternative with a simpler mathematical formulation (but likely with more biological complexity) for readers who wish to implement our circuit. We later demonstrate that this adjusted model can also display the same qualitative behaviours.

## 3. Results

The numerical simulations were conducted in MATLAB R2020b. As the ODE systems are too complex to be analysed in terms of the numerical stability [66,67], we first used the Runge–Kutta fourth-order method [68] to obtain a benchmark result for the models; for the purpose of efficiency, we used the forward Euler method with time step Δt=0.01 to carry out the entire analysis. We gradually reduced the time step from Δt=0.1, and the result remained unchanged, which demonstrates that the numerical method was stable in this case.

Figure 3a displays one simulation result for Fernando’s model (Equation (Equation 2)), and the parameter values used in the simulation are listed in Table 2. Here, we used the Hill coefficients recommended in [53], which are a=4 and b=2.

The first spike in *p* was stimulated by unconditioned stimulus $u_{1}$. The second (small) spike in response *p* was triggered by conditioned stimulus $u_{2}$. Of course, response *p* could be adjusted to 0 only in the presence of $u_{2}$ by setting the initial concentration of $\omega_{2}$ to 0 just as in the original paper. Here, we highlight the fact that various levels of the conditioned response are available to be chosen. The first conditioning was formed at the third spike, and the second conditioning was formed at the fifth spike in *p* when both $u_{1}$ and $u_{2}$ were present. The fourth and the sixth spikes in *p* represent the first and the second learned responses, respectively, when solely $u_{2}$ is present. The learned response was reinforced after repeated conditioning. Next, we move on to the discussion of how the two stimuli dissociate, namely, how the learned response is attenuated in the presence of consecutive conditioned stimuli alone after the formation of conditioning. The conjecture in the previous section was validated by the last four spikes in response *p* in Figure 3a. As is apparent, the sixth, seventh, and eighth spikes were of the same amplitude, as the time intervals were not wide enough. Conversely, the response started to decrease (shown by the last two spikes) when the time interval was further widened. This could be deemed to be a limitation for the model because, in some applications (e.g., immune inflammation), we may hope to force the stimuli to dissociate by the repeated cuing of the conditioned stimulus alone in a short time window after the formation of associative learning, which is one of the motivations for our novel design.

Furthermore, we study whether the qualitative behaviours of associative learning that we previously introduced were preserved or not apart from using the Hill coefficients recommended in [53]. The values for the other parameters remained the same, as shown in Table 2.

First, we fixed b=2 and altered the Hill coefficient for the weight molecules from a=1 to a=4. The respective responses of molecule *p* are displayed in Figure 4. As is obvious from the figure, the qualitative behaviours barely changed irrespective of the value of *a*, apart from the fact a=1 gave rise to a relatively notable response when only conditioned stimulus $u_{2}$ was present prior to conditioning. However, the result of a=1 could still be classified as a valid associative learning in broad terms, as the response triggered by the conditioned stimulus $u_{2}$ alone was more significant after conditioning than before.

Then, we studied the case when a=b, and we altered *a* from a=1 to a=4. The respective responses of molecule *p* are displayed in Figure 5, which shows that only a=1 gave rise to undesirable behaviours, as the concentration of response *p* never came down to 0. This is because a=1 leads to a large transient growth rate of the weight for conditioned stimulus $\omega_{2}$.

In all, Fernando’s model is robust to the variation in Hill coefficients even without exploring the other parameters. It may offer more flexibility to synthetic biologists, since two dimers are not required to be bound cooperatively for weight molecules $\omega_{1}$ and $\omega_{2}$. As we show in Figure 4, even a=1 and b=2 could produce desirable results, which may reduce the experimental complexity.

Figure 3b displays the simulation result for the model with forced dissociation (Equation (Equation 4)) using a=2, and the other parameter values used in the simulation are listed in Table 3.

The first spike in response *y* was stimulated by unconditioned stimulus *z*, and the second spike was stimulated by conditioned stimulus *x*. The first conditioning was formed at the third spike, and the second conditioning was formed at the fifth spike in *p* when both *z* and *x* were present. The fourth and sixth spikes in *p* represent the first and the second learned responses, respectively. The learned response was reinforced after repeated conditioning. As opposed to Fernando’s model, this model could successfully repress the learned response to the preconditioned level by means of the repeated cuing of the conditioned stimulus within a short time window, which was corroborated by the last two spikes in response *y* in Figure 3b.

Considering that our model is not as heuristic as Fernando’s model, it was necessary to validate that the reinforcement effect was indeed the result of the conditioning. Therefore, we removed the second and the third unconditioned stimuli *z* from the system, and the result is shown in Figure 6a. As can be seen, the reinforcement effect no longer existed without the conditioning of *x* and *z*. The result also validates our conjecture in the previous section that the presence of *x* and *z* speeds up the degradation of *u*, which is a game changer for the formation of the reinforcement effect.

Similarly to what we performed for Fernando’s model, we studied the behaviours of this model under various Hill coefficients *a* while keeping the other parameters specified in Table 3 unchanged.

From a=1 to a=4, the respective responses of molecule *y* are displayed in Figure 7. As can be observed, only a=2 and a=3 yielded desirable associative learning behaviours. The effect of conditioning was not discriminative for a=1, and for a=4, the response during conditioning was not more significant than the one when only the unconditioned stimulus is present.

The previously introduced adjusted model (Equation (Equation 5)) could also display the qualitative behaviours that are shown in Figure 6b when using a=2. The parameter values used in this simulation are listed in Table 4. The adjusted model could give rise to a more abundant response on the dimensionless scale compared to the original version (Figure 3b and Figure 6b).

Again, we studied the behaviours of the model with various Hill coefficients *a* without changing the other parameters specified in Table 4. As is apparent from Figure 8, the qualitative behaviours of the associative learning were largely preserved. However, the learned responses for cases a=3 and a=4 were less significant than that of a=2.

## 4. Discussion

In this work, we presented a detailed analysis of two advanced behaviours (reinforcement effect and forced dissociation) in associative learning. Fernando’s model could successfully demonstrate the reinforcement effect if we properly chose the parameter values. However, the attenuation of the learned response only occurred when the time interval between the learned responses was large enough in the sense that there was no way to force the learned response to decrease within a short time window. The model introduced in Section 2.2 could manifest a reinforcement effect and forced dissociation with the parameter values listed in Table 3, which could potentially provide more possibilities for the biological and medical applications of synthetic biology.

Having highlighted the contribution of our model, we must point out that it comes with a few constraints of which synthetic biologists need to be aware. First, the overall qualitative behaviour of the system is not very robust to the parameters aside from the Hill coefficients. We found that a 25% change in parameter values could lead to less desirable behaviours. An example is given in Figure 9, where $\alpha_{ux}$ was changed from 0.6 to 0.45. The first spike was then of the same amplitude as that of the third in response *y*. However, the stringent constraint can be relaxed if we expect to implement only one of the two behaviours, either reinforcement effect or forced dissociation. Second, it is extremely difficult if not impossible to control the response to the conditioned stimulus prior to conditioning at an insignificant level (second spike in *y* in Figure 3b) while maintaining the behaviours of interest. Hence, the model proposed here may not be a suitable candidate to emulate the behaviours of Pavlov’s dog, but fits the context of associative learning in a broader sense where conditioning and learning are major concerns. Third, the model demands that *v* remains abundant in order for associative learning to happen. Therefore, *v* needs to be supplemented to a sufficient level before the start of each experiment. Otherwise, conditioned stimulus *x* alone could result in an overexpression of the response.

Last but not least, we also want to mention several potential applications of our work in the field of synthetic biology and medicine. First, in the treatment for diabetes, Ye et al. [69] built a synthetic signaling cascade that enhances blood-glucose homeostasis. The reinforcement effect that we demonstrated in the models may pave the way for further adjustment to the circuit in the hope to attain the more efficient control of glucose level. Second, in the treatment of immune-mediated diseases, adoptive T-cell transfer technology shows immense promise in the treatment of immune-mediated disease such as cancer immunotherapy [70]. The feature of forced dissociation displayed in our model may provide more flexibility for experimentalists to shut off an excessive immune response once the tumours are eliminated. Lastly, some recent studies were focused on modelling the network of neurodegenerative markers [71,72]. The models that we discussed in this article may shed light on how to model these new findings at the genetic circuit level and build hierarchical neuronal architectures. The models may also play a supportive role in the existing technology that controls neurotransmitter release [73].

## Figures and Tables

**Figure 1 sensors-22-05907-f001:**
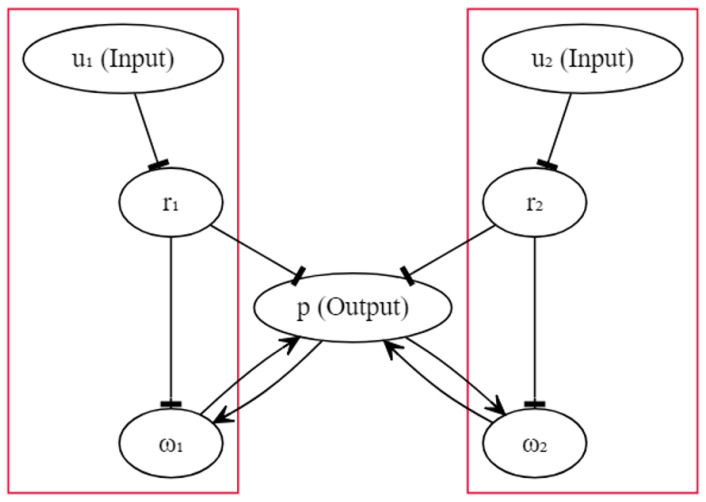
The schematic circuit of Fernando’s model.

**Figure 2 sensors-22-05907-f002:**
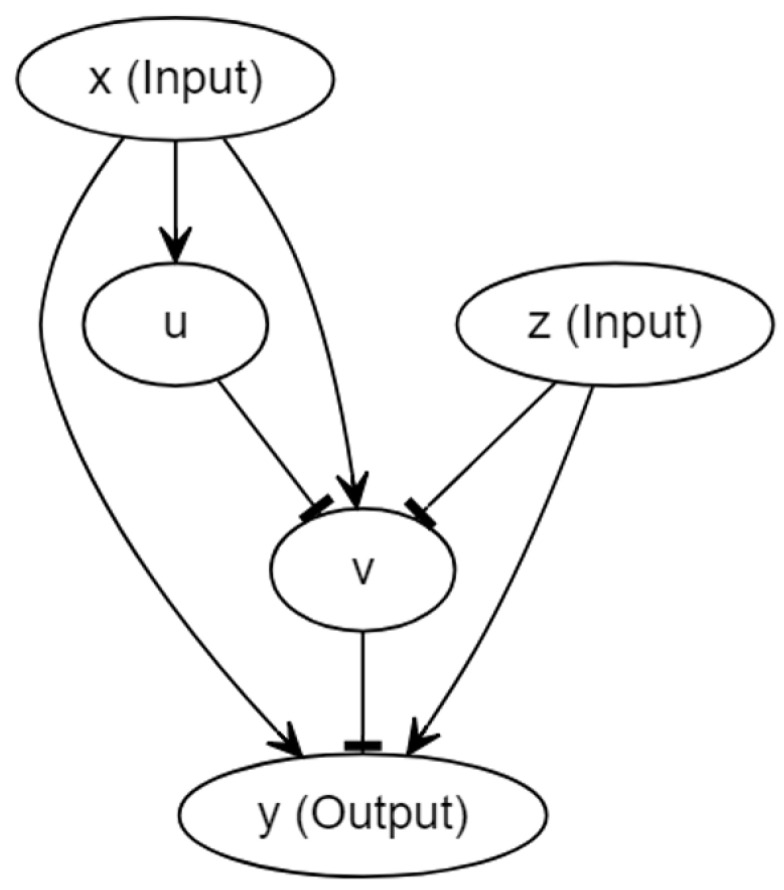
The schematic circuit of the model with forced dissociation.

**Figure 3 sensors-22-05907-f003:**
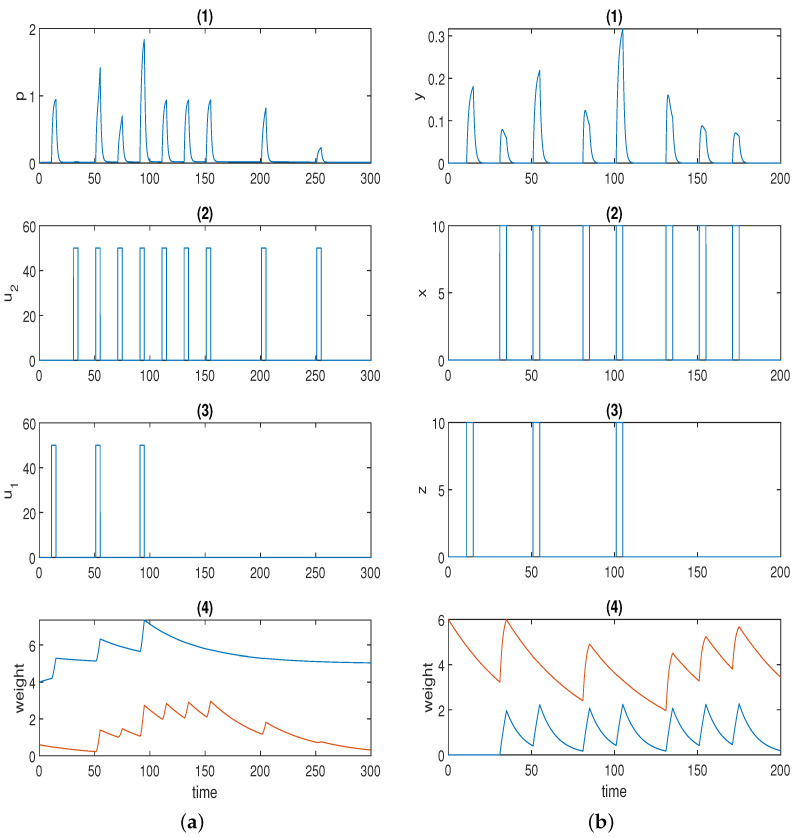
(**a**) Time series for Fernando’s model: (1) response molecule *p*; (2), (3) conditioned stimulus $u_{2}$ and unconditioned stimulus $u_{1}$; (4) weight molecule $\omega_{1}$ (blue) and $\omega_{2}$ (red); α=1, β=0.8, θ=0.02, τ=0.1, S=10. (**b**) Time series for the model with forced dissociation (1) response molecule *y*; (2), (3) conditioned stimulus *x* and unconditioned stimulus *z*; (4) weight molecule *u* (blue) and *v* (red); *a* = 2, *α_yx_* = 2, *α_yz_* = 4, *α_xyz_* = 4, *α_ux_* = 0.6, *α_vx_* = 1.5, *β_u_* = 0.1, *β_v_* = 0.02. In (1), the first spike was stimulated by the unconditioned stimulus, and the second spike was triggered by the conditioned stimulus; the first and second conditionings were formed at the third and fifth spikes, respectively; the first and the second learned responses were reflected by the fourth and the sixth spikes; the remaining spikes demonstrate whether the forced dissociation could be realised or not.

**Figure 4 sensors-22-05907-f004:**
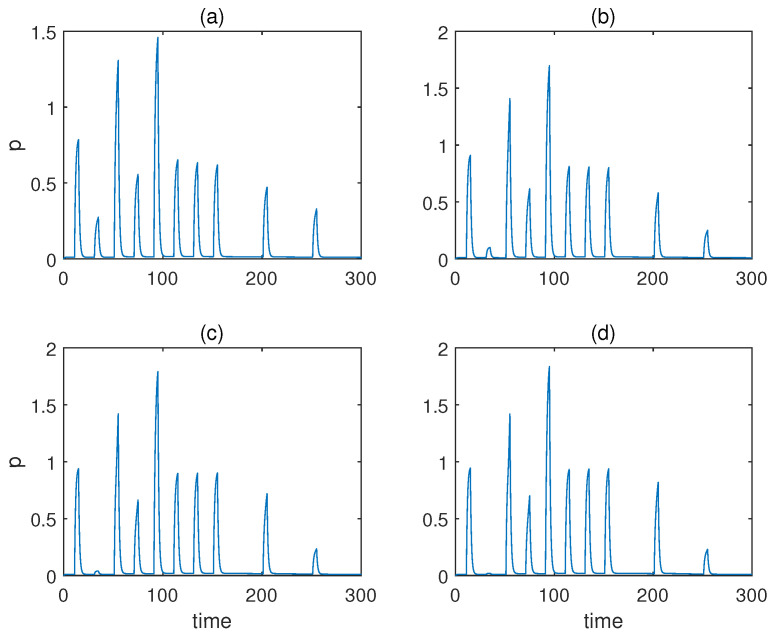
The concentration of molecule *p* under various Hill coefficients *a* when the value of *b* is fixed for Fernando’s model. (**a**–**d**) Results when *a* = 1, 2, 3, 4. *b* = 2, *α* = 1, *β* = 0.8, *θ* = 0.02, *τ* = 0.1, *S* = 10, respectively.

**Figure 5 sensors-22-05907-f005:**
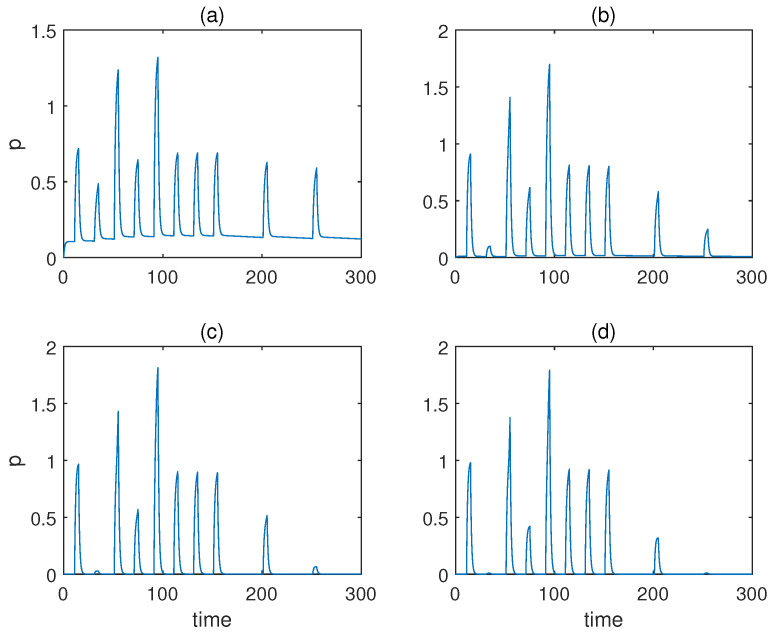
The concentration of molecule *p* under various Hill coefficients when a=b for Fernando’s model. (**a**–**d**) Results when *a* = 1, 2, 3, 4. *α* = 1, *β* = 0.8, *θ* = 0.02, *τ* = 0.1, *S* = 10, respectively.

**Figure 6 sensors-22-05907-f006:**
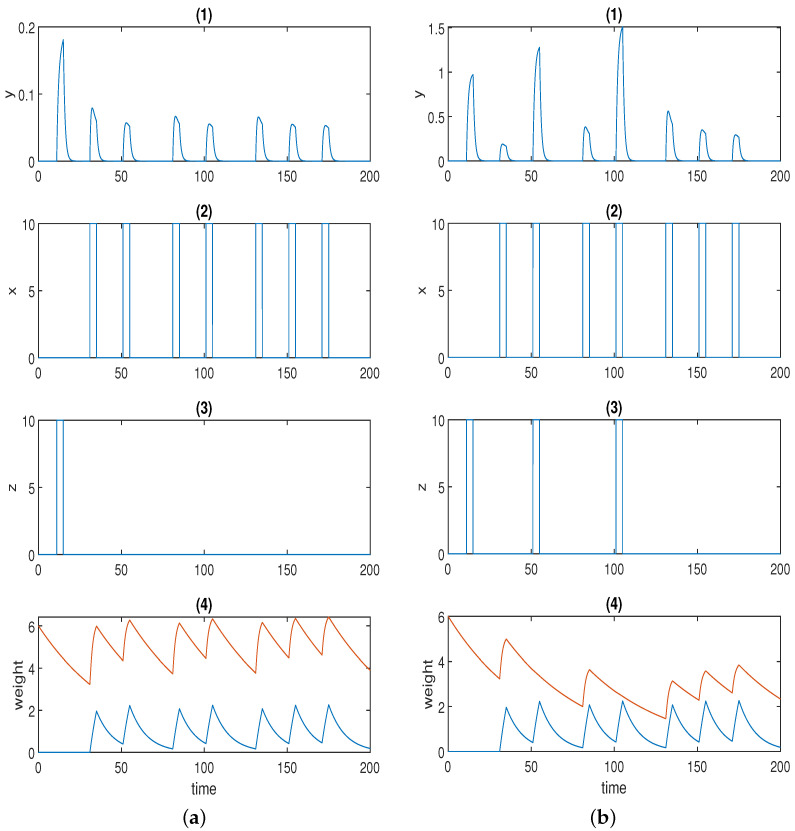
(**a**): Time series for the model with forced dissociation without conditioning (1) response molecule *y*; (2), (3) conditioned stimulus *x* and unconditioned stimulus *z*; (4) weight molecule *u* (blue) and *v* (red); *a* = 2, *α_yx_* = 2, *α_yz_* = 4, *α_xyz_* = 4, *α_ux_* = 0.6, *α_vx_* = 1.5, *β_u_* = 0.1, *β_v_* = 0.02. (**b**) Time series for the adjusted model with forced dissociation (1) response molecule *y*; (2), (3) conditioned stimulus *x* and unconditioned stimulus *z*; (4) weight molecule *u* (blue) and *v* (red); *a* = 2, *α_yx_* = 4, *α_yz_* = 1, *α_ux_* = 0.6, *α_vx_* = 1, *β_u_* = 0.1, *β_v_* = 0.02.

**Figure 7 sensors-22-05907-f007:**
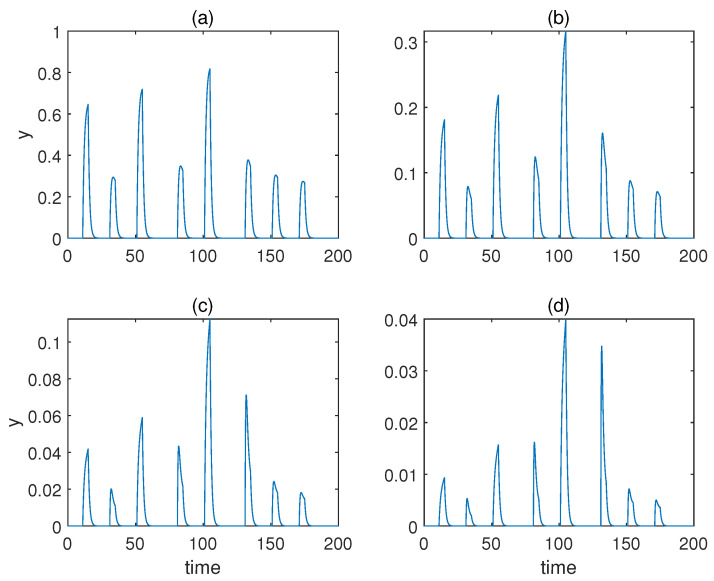
The concentration of molecule *y* under various Hill coefficients *a* for the model with forced dissociation. (**a**–**d**) Results when a=1,2,3,4. $\alpha_{yx}=2,\;\alpha_{yz}=4,\;\alpha_{xyz}=4,$
$\alpha_{ux}=0.6,\;\alpha_{vx}=1.5,\;\beta_{u}=0.1,\;\beta_{v}=0.02$.

**Figure 8 sensors-22-05907-f008:**
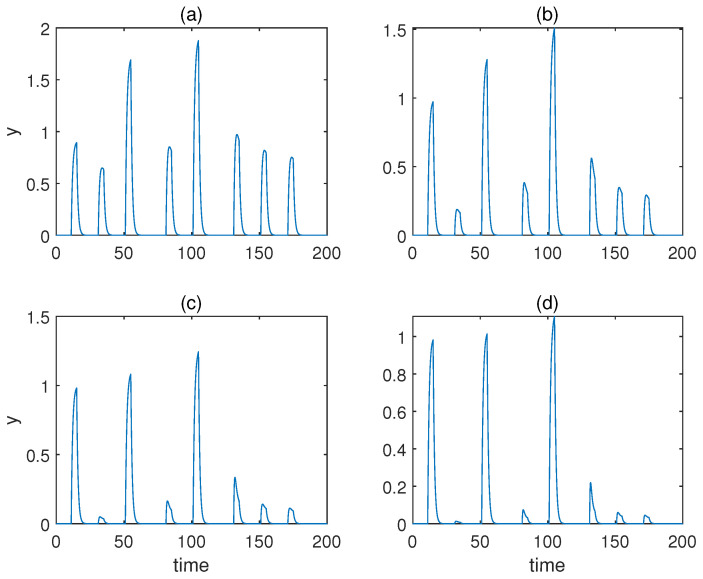
The concentration of molecule *y* under various Hill coefficients *a* for the adjusted model with forced dissociation. (**a**–**d**) Results when a=1,2,3,4. $\alpha_{yx}=4,\;\alpha_{yz}=1,$
$\alpha_{ux}=0.6,\;\alpha_{vx}=1,\;\beta_{u}=0.1,\;\beta_{v}=0.02$.

**Figure 9 sensors-22-05907-f009:**
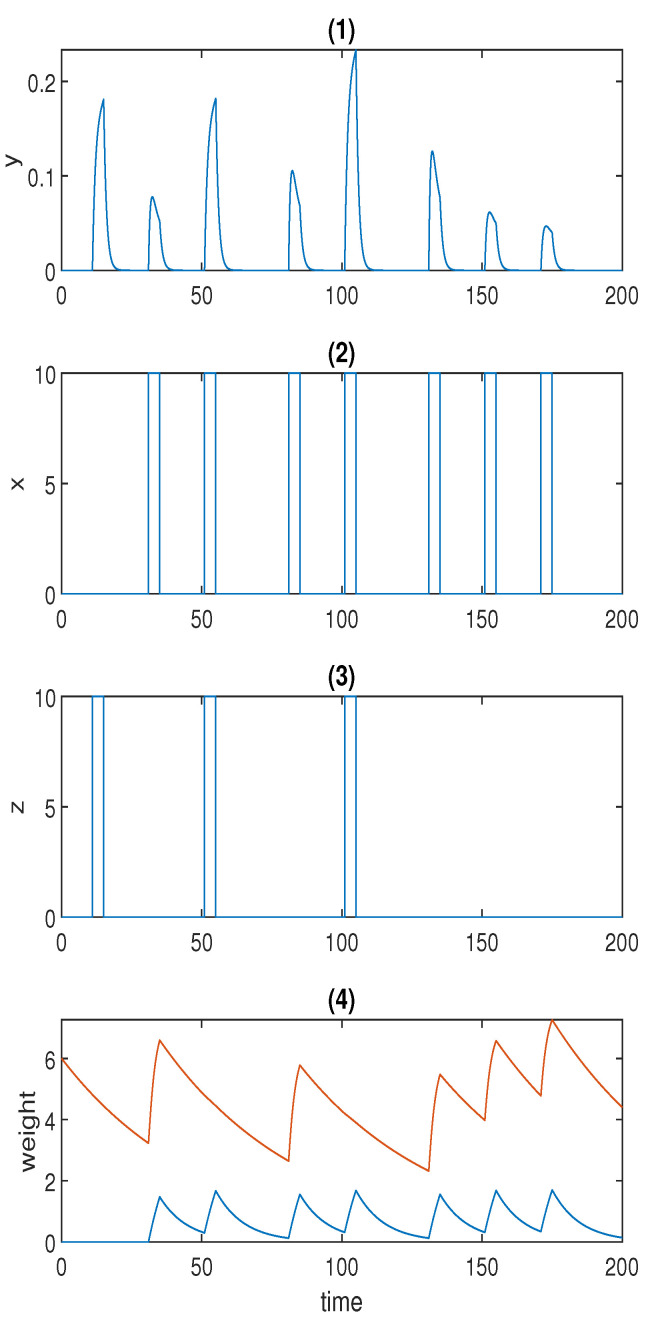
An example to illustrate the effect of change in parameter values on the model with forced dissociation. $a=2,\;\alpha_{yx}=2,\;\alpha_{yz}=4,\;\alpha_{xyz}=4,\;\alpha_{ux}=0.45,\;\alpha_{vx}=1.5,\;\beta_{u}=0.1,\;\beta_{v}=0.02$.

**Table 2 sensors-22-05907-t002:** Parameter values used in the simulation for Fernando’s model (Equation (Equation 2)).

Parameter	Value
α	1
β	0.8
θ	0.02
τ	0.1
*S*	10

**Table 3 sensors-22-05907-t003:** Parameter values used in the simulation for the model with forced dissociation (Equation (Equation 4)).

Parameter	Value
$\alpha_{yx}$	2
$\alpha_{yz}$	4
$\alpha_{xyz}$	4
$\alpha_{ux}$	0.6
$\alpha_{vx}$	1.5
$\beta_{u}$	0.1
$\beta_{v}$	0.02

**Table 4 sensors-22-05907-t004:** Parameter values used in the simulation for the adjusted model with forced dissociation (Equation (Equation 5)).

Parameter	Value
$\alpha_{yx}$	4
$\alpha_{yz}$	1
$\alpha_{ux}$	0.6
$\alpha_{vx}$	1
$\beta_{u}$	0.1
$\beta_{v}$	0.02

## Data Availability

https://github.com/zonglunli7515/Sensors-Associative-Learning-Reinforcement-Effect-and-Forced-Dissociation.

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
