# Peer review of "An Account of Models of Molecular Circuits for Associative Learning with Reinforcement Effect and Forced Dissociation"

_sensors, 2022, doi:10.3390/s22155907_

Round 1
Reviewer 1 Report
In general, this paper is very interesting with an innovative topic. The writing style is high and the manuscript is well structured and well organized.
I suggest to add statistical tabulated values, which can further elaborate on the analysis done in this paper.
In Fernando’s model, proper justification is required for choosing the parameter values.
In addition, a table containing comparisions among state-of-the-art techniques would further increse the understanding of readership.
Reviewer 2 Report
The manuscript studies continuous mathematical models of molecular circuits capable of robustly demonstrating associative learning, formulating a new model exhibiting reinforcement and forced dissociation that were often missing in prior models. The novel circuit design has relatively few parameters, intuitive interpretability, and robustness to certain key parameter variations.
The paper is well organized and appears scientifically sound. Figures and tables are well selected and detailed. The discussion section highlights potential model shortcomings as well as important biological implications. I have a few minor suggestions for improvement:
-The results section is a single relatively long section. For organizational purposes and readability, it may be helpful to include several subsections.
-The axis label fonts in several figures appear stretched out and slightly unclear (for example fig 7).
- There are some minor grammatical issues and typos throughout that should be carefully addressed:
Ex: Pg 1: “Over the past few decades, synthetic biology has witnessed rapid revolution in biotechnology industry and opened”
Pg 7: “However, the result of a = 1 can still be classified as a valid associative learning in broad term as the”
Pg 9: “As can be seen, the reinforcement effect no longer exists without the the conditioning of x and z.”
Pg 9: “The effect of conditioning is not discriminative for a = 1, and for a = 4, the response during condition is not more significant than the one when only unconditioned stimulus is present.”
